# A Retrospective Cohort Study of Oral Leukoplakia in Female Patients—Analysis of Risk Factors Related to Treatment Outcomes

**DOI:** 10.3390/ijerph18168319

**Published:** 2021-08-06

**Authors:** Shih-Wei Yang, Yun-Shien Lee, Pei-Wen Wu, Liang-Che Chang, Cheng-Cheng Hwang

**Affiliations:** 1Department of Otolaryngology-Head and Neck Surgery, Chang Gung Memorial Hospital, Keelung 204, Taiwan; a9665@cgmh.org.tw; 2College of Medicine, Chang Gung University, Taoyuan 333, Taiwan; lc2008@cgmh.org.tw (L.-C.C.); rrooyyaall@yahoo.com (C.-C.H.); 3Department of Otolaryngology-Head and Neck Surgery, New Taipei Municipal Tucheng Hospital, New Taipei City 236, Taiwan; 4Department of Biotechnology, Ming Chuan University, Taoyuan 333, Taiwan; bojack@mail.mcu.edu.tw; 5Genomic Medicine Research Core Laboratory, Chang Gung Memorial Hospital, Taoyuan 333, Taiwan; 6Department of Pathology, Chang Gung Memorial Hospital, Keelung 204, Taiwan

**Keywords:** female, leukoplakia, transformation, squamous cell carcinoma

## Abstract

Background: The aim of this study was to make a comparison of clinicopathological characteristics of oral leukoplakia between male and female patients following carbon dioxide laser excision for oral leukoplakia and analyze the factors associated with the treatment outcomes in female patients. Methods: Medical records of patients with oral leukoplakia receiving laser surgery from 2002 to 2020 were retrospectively reviewed and analyzed statistically. Results: A total of 485 patients were enrolled, including 412 male (84.95%) and 73 female (15.05%). Regarding the locations, the predilection site of oral leukoplakia in male patients was buccal mucosa (*p* = 0.0001) and that for women patients was tongue (*p* = 0.033). The differences of recurrence and malignant transformation between both sexes were not significant (*p* > 0.05). Among female patients, area of oral leukoplakia was the risk factor related to recurrence (*p* < 0.05). Clinical morphology and postoperative recurrence were the risk factors related to malignant transformation (*p* < 0.05). Conclusions: In comparison with male patients, there was no significant difference of the postoperative recurrence and malignant transformation of oral leukoplakia in female patients. Among the female patients, clinicians should pay more attention to large-sized and non-homogeneous leukoplakia, and postoperative recurrent lesions.

## 1. Introduction

Oral cancers are the sixth most common malignancy across the globe, and oral squamous cell carcinoma (OSCC) accounts for more than 90% of cases. More than 400,000 new cases of oral cancers were estimated to be diagnosed worldwide [1]. Although there is a wide variation in the incidence and mortality rate of OSCC in the different regions, the incidence of oral cancers remains high in the South and Southeast Asia [1,2]. OSCC is believed to be preceded by oral potentially malignant disorders (OPMDs), which are oral mucosa lesions with an increased risk of development of squamous cell carcinoma [3,4]. OPMDs include a variety of oral lesions, such as oral leukoplakia (OL), erythroplakia, submucous fibrosis, lichen planus, oral lichenoid reactions, and dyskeratosis congenita and epidemolysis bullosa [5,6,7]. Among the OPMDs, OL is the most common type of OPMD [5,8], and its prevalence is estimated about 4.11% of general population globally [9]. When the mucosa lesions look like morphological appearances of OL, the occurrence of carcinoma could be as high as 12.9% [10] and the consequential development of malignant change of those non-carcinomatous OL remains a challenging condition clinically. Regarding the analysis of OL undergoing malignant transformation, old age, female gender, area of leukoplakia exceeding 200 mm^2^, non-homogeneous type, and higher grades of dysplasia were reported to be risk factors [11,12]. Although female gender has been reported as an associated factor related to transformation to carcinoma [13,14,15,16,17,18], there were opposite perspectives on the role of female gender as a risk factor related to malignant transformation of OL [19]. The null hypothesis is that there is no difference of risk of malignant transformation of OL between the female patients and male patients. The first aim of this study was to make a comparison of the clinicopathological characteristics between the male and female patients who received carbon dioxide laser (CO_2_) excision for OL and the second aim was to conduct a deliberation on the group of female patients with OL to discern the treatment outcomes.

## 2. Materials and Methods

This study was approved by the Institutional Review Board of Chang Gung Memorial Hospital (License No.: 202100765B0) for ethical human research. Medical records of patients with OL that received transoral laser excision at the Department of Otolaryngology-Head and Neck Surgery of Keelung Chang Gung Memorial Hospital, from September 2002 to September 2020, were retrospectively reviewed. Every patient received thorough oral cavity examination by an otolaryngologist. Written informed consent was signed by the patient before surgical intervention. Incision biopsies were performed with blades for the large and non-homogeneous OL before CO_2_ laser excision was performed in some cases because some patients preferred to make a decision whether to receive a total excision after the pathology of a biopsy was available, while some patients underwent laser excision of the whole lesions directly. To avoid sampling errors between biopsies and excisions [20], only patients who received laser excision for OL were enrolled. Laser excision was conducted under local anesthesia. The CO_2_ laser used was UltraPulse^®^ Encore™ (Lumenis Inc., Yokneam, Israel), which broke down in 2015. Another laser machine of the same type was purchased in the same year. The power was set in a continuous wave mode with 12–15 Watt. Excisions were performed using a hand-held delivery device, the spot size being adjusted to 1 mm in diameter. A helium–neon aiming beam was provided to facilitate the guidance to the target lesion. Initially, the laser in a continuous-wave mode was set to outline the resection margin, situated at least 3 mm outside the apparent clinical margin of the targeted lesion. The depth of the excised mucosa was approximately 3–5 mm. The margin and depth were less when resecting hard palate or gingiva mucosa lesions. After the excision had been finished, the laser was shifted to defocus with the spot size being 2 mm in diameter and the outer 2 mm of the peripheral margins were vaporized to eliminate residual disease and to facilitate hemostasis. Excised areas healed by secondary intention, and reepithelialization was usually completed within 4–6 weeks [21,22,23]. Before surgery, the morphology of leukoplakia, including homogeneous and non-homogeneous [7,24], were first evaluated and photographed by the author (S.-W.Y.). Homogeneous OL is a uniformly flat, smooth, and whitish surface on the oral mucosa with or without fine cracks or fissures. Non-homogeneous OL can be further divided into three subtypes, including speckled, nodular, and verruciform types [5,7]. Speckled type leukoplakia, also termed erythroleukoplakia, contains a mixture of red and white areas but retains predominantly white coloration. Nodular leukoplakia usually presents with small polypoid outgrowths, rounded, red, or white excrescences. Verruciform type leukoplakia may exhibit a wrinkled or corrugated surface appearance [5,7,25,26]. The pictures of the outlook were later reviewed by two otolaryngology specialists and a consensus on the clinical morphology was reached. The clinical features of erythroplakia are smooth, velvety, possibly with an irregular margin and granular surface, and with a sharp demarcation; the red patch usually is situated at a level 0.1–0.2 mm lower than the surrounding oral mucosa [27]. Erythroplakia is different from erythroleukoplakia (or speckled type leukoplakia) and patients with erythroplakia were not recruited in this study.

Proliferative verrucous leukoplakia (PVL) is a distinct type of OPMD with persistence, gradual expansion with or without fusion of whitish plaque foci, and resistance to treatment [28]. It is an aggressive form of leukoplakia with a malignant transformation rate close to 50%, which is relatively high compared with other types of OPMDs [28,29]. Therefore, cases of PVL were excluded in the present study. There is no current consensus on the diagnostic criteria of PVL. Four conceptual proposals have been published [30,31,32,33] since PVL was first reported by Hansen et al. in 1985 [33]. In addition, an up-to-date conceptual proposal and diagnostic criteria for PVL analysis from different clinicopathological perspectives was provided [28]. In this study, the pattern of recurrence had been carefully examined; lesions with an exophytic, papillary, and warty appearance resembling PVL were excluded. All the multifocal lesions of OL were synchronous.

All the specimens, including biopsied and/or excised leukoplakia, were sent to the department of pathology and examined by two pathology specialists. The inclusion criteria of this study consisted of a clinical diagnosis of leukoplakia in the oral cavity, including buccal, tongue, mouth floor, labial, gum, hard palate, and retromolar regions, treatment with CO_2_ laser, and patients being over 20 years of age. Patients’ with age younger than 20, other kinds of OPMDs except leukoplakia (such as submucous fibrosis, lichen planus, and erythroplakia), previous treatment of OPMDs at other medical facilities, past history of oral cancer or radiation therapy on the head and neck area, surgical margins involved by hyperkeratosis or dysplasia, no agreeable pathological diagnosis made by two pathologists, overt carcinoma on inspection or initial pathological diagnoses being carcinoma or other malignancies, obvious ulceration, papilloma with a gross papillary appearance, treatment with laser vaporization, or inadequate data, were excluded.

The history of betel quid chewing, alcohol drinking, and tobacco use were obtained by a detailed questionnaire filled out by the patient at the first visit to the outpatient department. Current cigarette smokers were those who smoked one cigarette or more per day for at least one year; ex-smokers were those who did not smoke in the past 30 days [34,35,36]. Current or regular drinkers were those who drank more than four days a week and ex-drinkers (or former drinkers) were those who did not drink in the 12 months preceding interview [36,37]. Habitual betel quid chewers were those who chewed one quid or more daily for at least one year; ex-chewers were defined as having quit chewing betel quid for 6 weeks or longer [36,38]. The area of the leukoplakia was obtained from the measurement of the excised specimen on the pathological reports.

Pathological epithelial dysplasia was defined according to the WHO 2005 classification [39]. Postoperative recurrence was defined as a lesion of OL regrowing on the operated site after there was no evidence of OL for a definite period [40]. “Multifocal condition” described leukoplakia involving more than one part of the oral cavity mucosa. The area of OL in a patient was a summation of all leukoplakia lesions if more than one lesion occurred. When the patient had more than one lesion, the highest degree of pathology and most severe form of morphology were documented for analysis and statistical calculation on a per capita basis. The overall cumulative malignant transformation rate of OL was the case number of malignant transformation divided by the total case number of OL. Annual transformation rate (ATR) was the value of overall cumulative transformation rate divided by the average time (year) of OL cases developing carcinoma [11].

All surgical procedures were conducted by a defined surgical protocol under local anesthesia [22,23]. The patients’ postoperative follow-up courses were uneventful. All the patients were able to come back to clinic as scheduled without major complications or morbidities such as wound infections causing systemic septicemia, massive hemorrhage, localized paresthesia, and change of taste sensations. A small number of patients’ oral mucosa became more sensitive to spicy food in the short term. The phenomenon soon subsided.

### Statistical Analysis

Results are presented descriptively, with factors related to postoperative recurrence and malignant transformation of OL. For univariate analysis, the Fisher’s exact test, and one way analysis of variance between groups were performed. The survival analyses were made using Kaplan–Meier curves with log rank tests (for factor with two groups of subjects) and logistic regression model (for continuous variable such as body mass index, area of leukoplakia, or combined calculations of factors). Odds ratio (OR), hazard ratio (HR), and 95% confidence intervals (CIs) were calculated using a two-tailed test of significance (*p* < 0.05) for each factor. We followed the following parameters: (1) when the 95% CI did not include 1.0, the resulting OR (or HR) of the risk factor was statistically significant; (2) if the value of the OR (or HR) was greater than 1.0, the risk was increased, and (3) if the value was less than 1.0, the risk was reduced or protective. The Fisher’s exact tests were calculated using the MATLAB version R2015a (Mathworks Inc., Natick, MA, USA). Kaplan–Meier curves with log rank tests and multivariate Cox regression analysis model using the Statistical Package SPSS version 22 (SPSS Inc., Chicago, IL, USA) were used to determine the distinct factors affecting postoperative recurrence and malignant transformation of OL treated with CO_2_ laser.

The sample size and power estimate for two groups survival analysis for the factor area was according to the method described by Schoenfeld [41] and calculated with an online tool (https://sample-size.net/sample-size-survival-analysis/, accessed on 19 July 2021). Receiver operating characteristic (ROC) curve and the area under the ROC curve were measures of how well the area of OL could distinguish between lesions with or without postoperative recurrence. The optimal prediction of the cutoff point of the area of OL was based on the Youden Index J [=maximum (sensitivity + specificity − 1)] [42]. The ROC curves were performed with SPSS and the predictions and diagnostic tests were as described by Simel et al. [43].

## 3. Results

Overall, 799 patients with 1809 OPMD lesions underwent CO_2_ laser surgery from 2002 to 2020. Excluding patients whose OPMDs were not OL, and those whose follow-up time was less than 6 months, 485 patients with 997 lesions of OL were enrolled (Figure 1). Among the 485, 412 were male (84.95%) whose age ranged from 23 to 83 years with a median 52.0 and average 52.38 ± 11.70, and 73 were female (15.05%) whose age ranged from 25 to 83 years with a median 53.0 and average 53.03 ± 11.87. The oral habits of cigarette smoking, alcohol drinking, and betel quid chewing were found significantly more in male patients than in female patients (*p* < 0.0001, Table 1).

Regarding the locations, OL of the tongue and floor of the mouth more frequently occurred in the women than in men (*p* = 0.033, Table 1) and OL of buccal and other parts of the oral cavity except the tongue and mouth floor was more often seen in men than in women (*p* = 0.0001, Table 1). The predilection site of OL in male patients was buccal mucosa (*p* = 0.0003, Table 1) and for women patients was the tongue (*p* = 0.0054, Table 1). With respect to the other clinicopathological factors, including age, body mass index, *Candida* infection, multifocal disease, diabetes mellitus, metformin treatment, pathology, area of OL, and follow-up time, there was no difference between the male and female genders. The average follow-up time for male patients was 5.39 ± 3.78, and that for female patients was 5.91 ± 4.67 years.

Regarding the treatment outcomes of both sexes, the postoperative recurrence rate for male and female genders was 30.34% and 32.88%, respectively. The overall cumulative malignant transformation rate was 6.31% for male patients and 8.22% for female patients. The ATR was 1.76% for men and 2.08% for women. As for the treatment outcomes of OL, such as postoperative recurrence, cumulative malignant transformation, and ATR, no statistical difference was shown between the patients of different genders (Table 2).

The pathological grading of OL in the different or recurrent sites of a single patient may vary. It is not possible to correlate every patient with a single pathological result unless the patient had only one lesion. Thus, the highest degrees of pathological severity were recorded on a per capita basis. The numbers of cases of pathologically squamous hyperplasia, mild dysplasia, moderate dysplasia, and severe dysplasia were 109, 192, 63, and 48 in the male patients and 25, 27, 11, and 10 in the female patients, respectively. If a binary classification was adopted [39], high-risk lesions (moderate dysplasia and severe dysplasia) were outnumbered by the low-risk lesions (squamous hyperplasia and mild dysplasia) both in male and female patients but there was no significant difference.

Concerning the outcome measure in the group of all 73 female patients with OL, the average age was 56.73 ± 12.19 and median was 58.0 years. In the univariate analysis of postoperative recurrence, Kaplan–Meier curves with log-rank tests were used. Area of OL was the only significant factor related to postoperative recurrence (*p* = 0.001, HR 4.12, CI 95% 1.82–9.34). Cox proportional regression analysis showed an area of OL was also the independent prognostic factor associated with recurrence; the mean area of the postoperative recurrent lesions (4.81 ± 4.18 cm^2^) was larger than that of non-recurrent (1.79 ± 2.40 cm^2^) lesions (*p* = 0.008, HR 1.24, CI 95% 1.06–1.45, Table 3). In comparison with the female patients, multifocal disease (*p* = 0.01, HR 2.19, CI 95% 1.39–3.47), pathology (*p* = 0.011, HR 1.32, CI 95% 1.07–1.64), and area of OL (*p* < 0.0001, HR 1.14, CI 95% 1.09–1.18) were the independent prognostic factors related to recurrence in male patients.

The cut-off point of the area of 1.755 cm^2^ was calculated by the Youden index, and showed the best predictive value for postoperative recurrence (sensitivity = 0.75, specificity = 0.69); the area under the curve was 0.758 (Figure 2). Other demographic and clinicopathological variables were all non-significant. The area under the ROC curve (AUC) is 0.758. The straight dashed line represents the ROC curve expected by chance only.

In the univariate and multivariate analyses of malignant development of laser-treated OL in women patients, Kaplan–Meier survival analysis with log-rank tests and Cox proportional regression analysis, clinical morphology (*p* = 0.03, HR 9.81, CI 95% 1.76–54.73, Table 4) and postoperative recurrence (*p* = 0.020, HR 11.92, CI 95% 2.07–68.67, Table 4) was the associated factor with malignant transformation. There was a trend between postoperative recurrence and malignant transformation in the Cox regression analysis model (*p* = 0.07, Table 4). Other variables, including age, body mass index, oral habits, *Candida* infection, multifocal disease, subsites, diabetes mellitus, metformin treatment, and pathology, were not statistically significant. In the male patients with OL, age (*p* = 0.005, HR 5.23, CI 95% 1.63–16.80), body mass index (*p* = 0.029, HR 0.36, CI 95% 0.15–0.90), betel quid chewing (*p* = 0.03, HR 10.90, CI 95% 1.25–94.79), location (*p* = 0.006, HR 0.91, CI 95% 0.017–0.50, for OL on buccal and other sites except tongue and mouth floor), multifocal disease (*p* = 0.002, HR 6.99, CI 95% 1.99–24.54), and pathology (*p* < 0.0001, HR 4.64, CI 95% 2.47–8.72) were independent prognostic factors related to postoperative malignant transformation.

## 4. Discussion

This retrospective study of female patients with OL demonstrated that the risk factor for recurrence after laser surgery was the size of the area, and the risk factor for malignant transformation was postoperative recurrence (Table 3 and Table 4). With regard to the clinicopathological characteristics, the location of OL in female patients more often occurred on the tongue subsite (*p* = 0.033, Table 1), which was different from male patients whose OL more often occurred on the buccal mucosa (*p* = 0.0001, Table 1). The difference of oral habits, including cigarette smoking, alcohol drinking, and betel quid chewing, between the female and male genders were also found; the prevalence of use of cigarette, alcohol, and betel quid was significantly higher in men (*p* < 0.0001, Table 1) and in the Cox proportional regression analysis, betel quid chewing (*p* = 0.03, HR 10.90, CI 95% 1.25–94.79) was a significant prognostic factor related to malignant transformation in male patients but not in the female patients. The habit of betel quid chewing was far less prevalent in females than in males and the impact to the male patients was observed. From the analysis of treatment outcomes, the time for developing OSCC from OL, postoperative recurrence rate, overall cumulative malignant transformation rate, and ATR were all non-significant despite the different predilection site of OL and different oral habits between the male and female genders (*p* > 0.05, Table 2). Based on these analysis data, the treatment outcomes did not differ in both sexes despite the disparity between some exogenous etiological factors. Accordingly, the inherent physiological sex differences may play part of the role in the responses to the treatment and pathogenesis of malignant change of OL, which is worthy of further research.

Clinical morphology of OL can be classified as homogeneous OL and non-homogeneous OL based on the texture, color, thickness, and regularity of the lesion [25]. Even there was variation in the study designs, treatment modalities among different studies, the risk of malignant transformation rate of non-homogeneous OL was higher than homogeneous type [7,44]. In a retrospective analysis of clinical features of oral cancer and OPMDs with and without oral epithelial dysplasia over a 12-year period, non-homogeneous mucosa lesions were found to be an independent indicator of dysplasia [45]. In the present study, the clinical morphology (homogeneous vs. non-homogeneous) was not a risk factor in the postoperative recurrence but non-homogeneous OL was a significant risk factor associated with the malignant transformation in the log-rank test in female patients; the non-homogeneous OL was shown to have 9.81-fold increased risk for malignant change compared with the homogeneous OL (*p* = 0.03, HR 9.81, CI 95% 1.76–54.73, Table 4). Clinical morphological outlook is an intuitive observation of OL and important information before surgical intervention. Clinicians should pay more attention to the non-homogeneous OL. Neither for recurrence nor for malignant transformation was clinical morphology a risk factor in male patients.

In a systematic review and meta-analysis of 49 studies on diabetes and oral cancer/OPMDs, patients with diabetes mellitus had a higher prevalence and greater chance for oral cancer and OPMD development [46,47]. Metformin was found to have beneficial effects on head and neck cancer risk and overall survival in a systematic review with meta-analysis of six studies on metformin [48]. However, diabetes and metformin were not risk factors for postoperative recurrence and malignant change of OL in female and male patients in the present study.

The etiology of OL is multifactorial and tobacco smoking, alcohol use, and betel quid chewing were found to be etiological factors for OL and OSCC [10,49,50]. Betel quid chewing was associated with higher chance of presence of *Candida* infection, which was thought to be an uncommon risk factor for OL and oral cancer [49]. Distinct clinicopathological differences were reported in single OL lesion and multiple OL lesions [51], and widespread multiple OL had a higher potential for malignant transformation [52]. In this series, oral habits, *Candida* infection, and multifocal lesions were not significant factors for postoperative recurrence and malignant change of OL in female patients. However, multifocal lesions of OL were an associated prognostic factor with recurrence (*p* = 0.01, HR 2.19, CI 95% 1.39–3.47) and malignant transformation in male patients (*p* = 0.002, HR 6.99, CI 95% 1.99–24.54).

Although excision could not achieve the goal of primary prevention or avoidance of malignant development of OL, the pathological examination of the excised specimen can provide very important information. In the present study, there were no major complications postoperatively, which showed that laser excision for OL was a safe treatment with low morbidity. Some patients preferred biopsies before total excisions for OL. However, the discrepancy between biopsies and excisions [20] should not be neglected. Different histopathological degree of severity of cytological and architectural abnormalities can be categorized as mild, moderate, or severe dysplasia [7,9,53]. Appearance of dysplasia is an ominous sign, and a higher degree of dysplasia is usually associated with higher risk of malignant transformation of OL [7]. Malignant transformation from pathological non-dysplastic leukoplakia may also occur [45,54]. For patients with presence of dysplasia in biopsies of OL, laser excision is suggested. For patients without dysplasia, when the clinical morphology changes, or the size of OL area increases, further biopsy or excision is also suggested. In the female patients, the trend of transformation in high-risk OL (moderate and severe dysplasia) was observed (*p* = 0.09, Table 4) but not statistically significant. Regarding postoperative recurrence, pathology was not significant. In the male patients, pathology was an independent prognostic factor related to recurrence (*p* < 0.0001, HR 4.64, CI 95% 2.47–8.72) and malignant transformation (*p* < 0.0001, HR 4.64, CI 95% 2.47–8.72). Judging from the results, pathology remained a critical risk factor in both sexes.

It is obvious that female patients are a relative minority in the group of patients with OSCC. Similarly, it could be found that the number of female patients with OL was far less than that of male patients in this cohort study (73 vs. 412). OL was more commonly seen in males than females, which was also found in the previous studies [9,12,22,23,55]. In this study, men were more exposed to smoking, drinking, and betel quid chewing than women (Table 1), which was the major reason why female patients were outnumbered by male patients and these oral habits were also risk factors for OL and OSCC [6,10,44]. Although the case number and incidence of OL in females were not as large as for males, the postoperative recurrence and development of malignant change of OL in female patients could be as serious as in male patients. From a gender perspective, “female” has been regarded as a risk factor for malignant transformation of OL in some studies [13,14,15,16,17,18], but “male” has not. Reviewing the literature about the research on OL, neither malignant transformation of OL focusing solely on female patients nor comparison of treatment outcomes between both genders has been reported so far. In the viewpoint of malignant transformation, the potential risk of malignant transformation of OL in female gender should not be overlooked and deserves the attention of clinicians.

The recurrence rate in this cohort of female patients with OL was 32.88%, which was similar to the previous studies of OL following laser surgery whose postoperative recurrent rate of was 7.7–38.1% [56,57]. In the Kaplan–Meier survival analysis, log-rank tests, and Cox proportional regression analysis of postoperative recurrence of OL in female patients, the size of OL area was the only associated factor with recurrence (Hazard ratio: 4.12, CI 95% 1.82–9.34, *p* = 0.001, Table 3) and also the only independent prognostic factor related to recurrence (Hazard ratio: 1.24, CI 95% 1.06–1.45, *p* = 0.008, Table 3). The mean area of the OL with postoperative recurrence (4.81 ± 4.18 cm^2^) was significantly larger than that of non-recurrent lesions (1.79 ± 2.40 cm^2^). Not only in the present study, size of area was also a significant factor associated with postoperative recurrence in the studies of oral tongue leukoplakia [58], elderly patients with OL [59], and patients with oral erythroplakia [21]. In addition to being a critical factor related to recurrence, the area of OL was reported to be a risk factor affiliated with malignant change of OL in several studies [11,60,61]. In a systemic review of 24 observational studies on malignant transformation of OL, the area exceeding 200 mm^2^ was one of the significant determinants contributing to malignant change [11]. In a retrospective cohort study of 144 patients with OL, a large-sized lesion (≥4 cm) was shown to be the only significant predictor of malignant transformation [60]. In our study, a cut-off point of 1.755 cm^2^ was calculated based on the Youden Index J with 0.758 of area under curve (Figure 2) to achieve the best predictive value for postoperative recurrence (Table 3). In contrast, area was also a significant prognostic factor related to recurrence (*p* < 0.0001, HR 1.14, CI 95% 1.09–1.18), but not a risk factor related to malignant change of OL in male patients (*p* = 0.14).

The predominantly affected sites of OL is likely related to the etiologic factors and therefore may be different by the geographic locations and local habits. For example, in Sudan Toombak, or a local type of smokeless tobacco, is excessively used by the habitants and the most common sites of leukoplakia are lip and gingiva [55], which are not frequently seen in India and Taiwan, areas endemic for betel chewing. Among betel quid chewers, buccal mucosa is likely to be the most affected site, whereas in those reverse smokers, the OL lesions more often occur on the palate [7]. The lateral border of tongue and the floor of the mouth are anatomically contiguous and the most common sites for OPMDs and OSCC in the developed world, where smoking of tobacco and alcohol consumption are the most important etiologic factors [7]. According to the important gender statistics database provided by Taiwan National Health Service, Ministry of Health and Welfare, the rate of smoking, betel nut chewing, and alcohol consumption among men and women in the general population over 18 in 2017 was 26.4% vs. 2.3%, 22.2% vs. 1.1%, and 53.4% vs. 33.0% (men vs. women) (https://www.gender.ey.gov.tw/gecdb/Stat_Statistics_Info.aspx, accessed on 13 May 2020). In our series, the case number of oral habits in female patients was significantly outnumbered by male patients. In addition, among female patients, there were significantly fewer people who have a habit of drinking and chewing betel nuts than of smoking (Table 1). These were compatible with the oral habits in general adult population. From a comparative clinical study in 16,379 Taiwanese patients with resected buccal and tongue squamous cell carcinoma whose date retrieved from Cancer Registry Database between 2004 and 2012, patients with buccal cancer had a higher prevalence of males (*p* < 0.0001) compared with tongue cancer. The phenomenon of male predominance in buccal cancer could be explained by the fact that betel quid chewers were generally males [62]. In a study of 112 cases of stage I oral tongue cancer in Baltimore, stage I tongue squamous cell carcinoma was found to be more common in women and was associated with pre-existing leukoplakia [63]. Why OL of buccal mucosa occurred more in male patients and OL of tongue more in female patients may not be able to be answered fully in the present study and the studies about the predilection subsites of OL between both genders were few in the literature. Oral habits might play a part in the role. These findings could also serve as a preliminary reference to future studies.

In this cohort, the overall cumulative malignant transformation rate and ATR of OL in female patients were 8.22% and 2.08%; while those in male patient were 6.31% and 1.76%, respectively. The ATRs in both genders were similar to the previous reports of OL whose ATRs ranged from 1.08% to 4.90% [15,16,58,60,64,65,66,67,68,69]. In the Kaplan–Meier survival analysis and log-rank tests, postoperative recurrence was the factor related to malignant change of OL (Hazard ratio: 11.92, CI 95% 2.07–68.67, *p* = 0.02, Table 4) after laser surgery. In the Cox proportional regression analysis model, postoperative recurrence was not significant but the trend towards malignant transformation of OL existed (*p* = 0.07, Table 4). Recurrence, defined as regeneration of whitish patch on the same site after surgical excision, implies treatment failure and was found to be a risk factor for malignant transformation of OL in the previous studies [23,58,70]. Usually, the extent of excision of the OL lesion was made based on the judgment from the surgeon’s naked eyes on the region of the lesion, the outline of excision was set about 3 mm outside the clinical margin of the targeted OL in order to achieve an adequate and total extirpation of OL [22,23]. According to the theory of field cancerization and some molecular biological studies, genetically altered epithelial cells can occur more widely than can be detected by the visual and pathological examination [20,52,57,71,72,73]. Large-sized OL means more disease burden. This may explain why there are postoperative recurrence and malignant transformation even after surgical extirpation or medical intervention on the OL lesions.

In the present study, survival analysis showed area of OL reached the statistical significance (α = 0.05). Then we estimated the sample size for survival analysis [41]. Based on a significant *p* value (α = 0.05) and power (0.8), at least 127 cases were needed for male patients and at least 60 cases were needed for female patients. Of the cases enrolled in this study, 412 were male and 73 were female patients. Therefore, the sample size was adequate.

There exist some limitations in this study. First, this is a retrospective chart-review cohort study, there was no control group and imbalance of case distribution in both sexes could not be avoided. Besides, some information of patients’ data was missing during the review of chart records, but the proportion was not large. Second, the sample size of female patients was relatively small. Further large-scale, prospective, multi-centered studies are needed to elucidate the disease entity in the group of female patients. Third, more in-depth studies at the level of genetic and molecular biology are warranted to unveil the differences between patients with OL of both sexes. Finally, the external validity of our findings is possibly a concern due to the differences of participants in ethnic and environmental factors among other parts of the world. Our results may suggest avenues for research in the future.

## 5. Conclusions

Apparently, the clinical characteristics and treatment outcomes in male and female patients with OL were different. We especially identified female patients as the subject of the research, hoping to find out something more about the female patients with OL, who are the minority in this field. The findings of our study revealed that there was no significant difference in the postoperative recurrence rate and malignant transformation rate of OL treated by CO_2_ laser excision between male and female patients. However, the risk factors related to recurrence and development of OSCC were different in both sexes. Among the female patients with OL receiving laser surgery, the area of OL was the risk factor and independent prognostic factor related to postoperative recurrence and the cut-off point was 1.755 cm^2^. The risk factors for the development of OSCC from treated OL were non-homogeneous OL and postoperative recurrence. It should be noted that medical practitioners have to be aware that although the cases of female patients with OL may not be as commonly seen as male patients, the risk of malignant transformation in female patients is not lower than male patients and should not be overlooked. Clinicians need to pay more attention to large-sized OL, non-homogeneous leukoplakia and postoperative recurrent lesions in female patients with OL.

## Figures and Tables

**Figure 1 ijerph-18-08319-f001:**
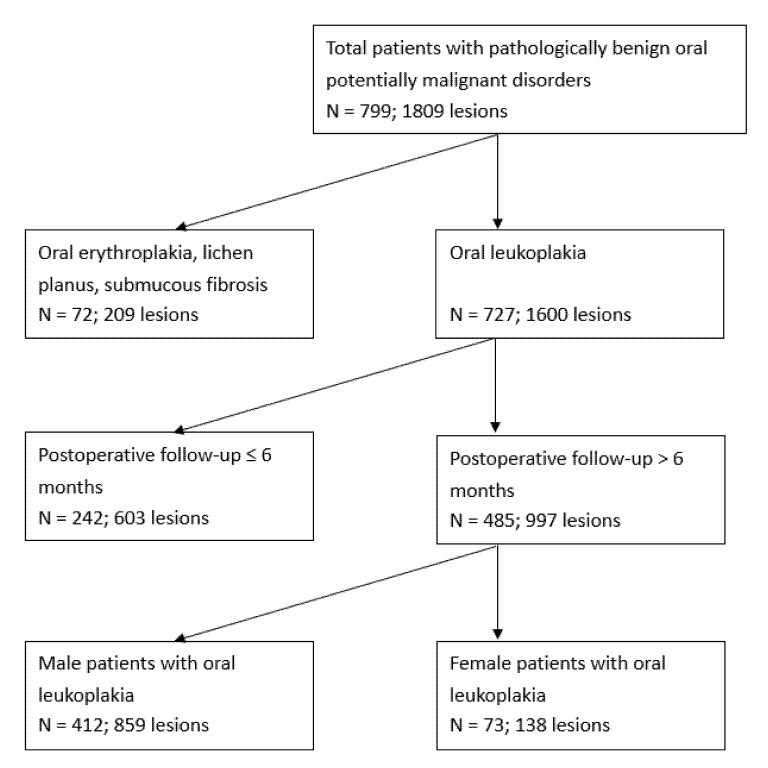
Flow chart of the cohort study demonstrating the number of participants under various screening criteria and conditions. Number of the patients (N) and lesions were shown. Since more than 1 lesion was recorded in patients with multifocal disease, the numbers of total lesions were more than the numbers of total patients.

**Figure 2 ijerph-18-08319-f002:**
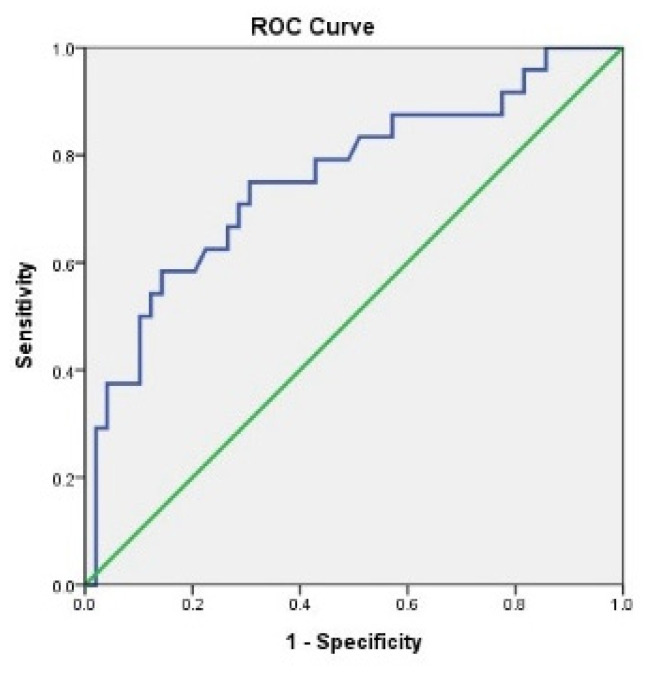
The receiver operating characteristic (ROC) curve analysis was used to predict postoperative recurrence. Each point on the ROC corresponds to a value on the area of oral leukoplakia.

**Table 1 ijerph-18-08319-t001:** The demographic data, and clinicopathological characteristics oral leukoplakia in male (n = 412) and female (n = 73) patients.

Factors		Male (n = 412)	Female (n = 73)	Odds Ratio	CI 95%	Fisher’s Exact *p* Value
Age (yr), mean ± stand deviation: 53.03 ± 11.87, median: 53.0						ns
	≤65	354	58	1.0		
	>65	58	15	1.58	0.84–2.97	
Body mass index *		25.86 ± 3.75	24.87 ± 4.92	0.93	0.87–1.0	ns **
Cigarette smoking †						<0.0001
	Non-smoker	62	47	1.0		
	Ex-smoker	154	9	0.077	0.036–0.17	
	Current smoker	195	17	0.12	0.062–0.21	
Alcohol drinking ‡						<0.0001
	Non-drinker	227	66	1.0		
	Ex-drinker	107	6	0.19	0.081–0.46	
	Current drinker	77	1	0.045	0.0061–0.33	
Betel quid chewing §						<0.0001
	Non-chewer	171	66	1.0		
	Ex-chewer	211	5	0.061	0.024–0.16	
	Current chewer	29	2	0.18	0.042–0.77	
*Candida* infection						ns
	No	375	67	1.0		
	Yes	37	6	0.91	0.37–2.23	
Multifocal disease						ns
	No	272	54	1.0		
	Yes	140	19	0.68	0.39–1.20	
Location (Tongue and mouth floor)						0.033
	No	329	50	1.0		
	Yes	83	23	1.82	1.05–3.16	
Location (Buccal and other sites except tongue and mouth floor)						0.0001
	No	26	16	1.0		
	Yes	386	57	0.24	0.12–0.47	
Location: All subsites in the oral cavity ‖						
	Buccal	331	44	0.37	0.22–0.63	0.0003
	Tongue	89	27	2.13	1.25–3.62	0.0054
	Mouth floor	4	1	1.42	0.16–12.86	ns
	Retromolar	85	16	1.08	0.59–1.97	ns
	Hard palate	20	3	0.84	0.24–2.90	ns
	Gum	48	3	0.33	0.099–1.07	ns
	Labial	20	5	1.44	0.52–3.97	ns
Diabetes mellitus ¶						
	No	326	57	1.0		ns
	Yes	83	16	1.1	0.60–2.02	
Metformin treatment #						ns
	No	340	61	1.0		
	Yes	68	12	0.98	0.50–1.93	
Clinical morphology						ns
	Homogeneous	274	44	1.0		
	Non-homogeneous	138	29	1.31	0.78–2.18	
Pathology						ns
	Squamous hyperplasia	109	25	1.0		
	Mild dysplasia	192	27	0.61	0.34–1.11	
	Moderate dysplasia	63	11	0.76	0.35–1.65	
	Severe dysplasia	48	10	0.91	0.40–2.04	
Area of oral leukoplakia (cm^2^)		2.72 ± 3.29	2.78 ± 3.38	1.01	0.93–1.08	ns **
Follow-up time (year)		5.39 ± 3.78	5.91 ± 4.67	1.03	0.97–1.10	ns **

Abbreviation: OR, odds ratio; CI, confidence interval; ns, not significant. * Six missing data of body mass index (n = 479). † One missing data in the group of male patients with or without the habit of cigarette smoking (n = 411). ‡ One missing data in the group of male patients with or without the habit of alcohol drinking (n = 411). § One missing data in the group of male patients with or without the habit of betel quid chewing (n = 411). ‖ There could be more than one lesion on one subside, including the primary and recurrent lesions. The comparison was made in patients within versus outside the location. ¶ Three missing data in the group of male patients with or without diabetes mellitus (n = 409). # Four missing data in the group of male patients who took metformin or not (n = 408). ** Comparison was done with a logistic regression mode.

**Table 2 ijerph-18-08319-t002:** The demographic data, and clinicopathological characteristics oral leukoplakia in male (n = 412) and female (n = 73) patients.

Factors		Male (n = 412)	Female (n = 73)	OR	CI 95%	*p* Value	HR *	CI 95%	*p* Value
Postoperative recurrence									ns
	No	287	49				1.0		
	Yes	125	24				0.98	0.63–1.54	
Malignant transformation									ns
	No	386	67				1.0		
	Yes	26	6				1.07	0.41–2.75	
Postoperative recurrence rate (%)		30.34%	32.88%	1.12	0.66–1.91	ns			
Time period for development of malignant transformation (year)		3.58 ± 3.43	3.95 ± 3.07	1.03	0.80–1.33	ns			
Cumulative malignant transformation rate (%)		6.31%	8.22%	1.33	0.53–3.35	ns			
Annual transformation rate (%) †		1.76%	2.08%			ns			

Abbreviation: OR, odds ratio; CI, confidence interval; HR, hazard ratio; ns, not significant. * Hazard ratio was the evaluation of treatment outcome when postoperative recurrence and malignant transformation were analyzed with Kaplan–Meier survival analysis. † The annual transformation rate was calculated by the malignant transformation rate divided by the average time of development of carcinoma (year).

**Table 3 ijerph-18-08319-t003:** Log-rank tests and Cox proportional regression analysis of postoperative recurrence (n = 24) in female patients who received laser surgery for oral leukoplakia (n = 73).

Variable		Recurrence		Log-Rank Tests				Cox Proportional Regression Analysis			
					Confidence Interval 95%				Confidence Interval 95%		
		No (n = 49)	Yes (n = 24)	Hazard Ratio	Upper	Lower	*p* Value	Hazard Ratio	Upper	Lower	*p* Value
Age (yr), mean ± standard deviation: 56.73 ± 12.19, median: 58.0				1.39	0.47	4.14	0.75	1.01	0.96	1.05	0.78
	≤65	39	19								
	>65	10	5								
Body mass index *				1.19	0.52	2.70	0.84	1.00	0.996	1.002	0.55
	≤24	22	10								
	>24	24	13								
Cigarette smoking				1.09	0.47	2.55	0.99	0.50	0.13	1.94	0.32
	Non-smoker	32	15								
	Ex-smoker	7	2								
	Current smoker	10	7								
Alcohol drinking				2.07	0.55	7.85	0.47	1.24	0.042	36.34	0.90
	Non-drinker	46	20								
	Ex-drinker	3	3								
	Current drinker	0	1								
Betel quid chewing				1.44	0.43	4.81	0.78	1.29	0.05	33.01	0.88
	Non-chewer	46	20								
	Ex-chewer	2	3								
	Current chewer	1	1								
*Candida* infection				1.21	0.38	3.87	0.98	1.16	0.19	7.04	0.88
	No	47	20								
	Yes	2	4								
Multifocal disease				1.89	0.78	4.57	0.24	1.48	0.47	4.68	0.50
	Single lesion	40	14								
	Multiple sites of lesions	9	10								
Location (Tongue and mouth floor)				0.77	0.33	1.77	0.69	0.12	0.11	1.29	0.08
	No	34	16								
	Yes	15	8								
Location (Buccal and other sites except tongue and mouth floor)				1.82	0.74	4.47	0.28	0.14	0.10	2.00	0.15
	No	12	4								
	Yes	37	20								
Diabetes mellitus				1.75	0.65	4.77	0.4	1.46	0.81	26.28	0.80
	No	40	17								
	Yes	9	7								
Metformin treatment				1.68	0.58	4.82	0.49	2.87	0.13	47.32	0.55
	No	43	18								
	Yes	6	6								
Morphology				2.41	0.99	5.85	0.085	2.10	0.68	6.45	0.2
	Homogeneous	32	12								
	Non-homogeneous	17	12								
Pathology				1.67	0.70	4.00	0.35	0.65	0.20	2.15	0.48
	Low risk lesions †	38	14								
	High risk lesions ‡	11	10								
Area of oral leukoplakia (cut-off value: 1.755 cm^2^)				4.12	1.82	9.34	**0.001**	1.27	1.08	1.49	**0.04**
	≤1.755	34	6								
	>1.755	15	18								

Bold values denote statistically significant *p* value. * Four missing data of body mass index (n = 69). † Low risk lesions included squamous hyperplasia and mild dysplasia. ‡ High risk lesions included moderate and severe dysplasia.

**Table 4 ijerph-18-08319-t004:** Log-rank tests and Cox proportional regression analysis of postoperative malignant transformation (n = 6) in female patients who received laser surgery for oral leukoplakia (n = 73).

Variable		Malignant Transformation		Log-Rank Tests				Cox Proportional Regression Analysis			
					Confidence interval 95%				Confidence Interval 95%		
		No (n = 67)	Yes (n = 6)	Hazard Ratio	Upper	Lower	*p* Value	Hazard Ratio	Upper	Lower	*p* Value
Age (yr)				0.69	0.10	4.70	0.91	0.41	0.026	6.44	0.52
	≤65	53	5								
	>65	14	1								
Body mass index *				1.45	0.25	8.55	0.97	NA	NA	NA	NA
	≤24	30	2								
	>24	34	3								
Cigarette smoking				0.40	0.08	2.11	0.51	NA	NA	NA	NA
	Non-smoker	42	5								
	Ex-smoker	9	0								
	Current smoker	16	1								
Alcohol drinking				0.31	0.03	3.21	0.70	NA	NA	NA	NA
	Non-drinker	60	6								
	Ex-drinker	6	0								
	Current drinker	1	0								
Betel quid chewing				0.31	0.03	3.23	0.70	NA	NA	NA	NA
	Non-chewer	60	6								
	Ex-chewer	5	0								
	Current chewer	2	0								
*Candida* infection				1.75	0.14	22.47	0.83	NA	NA	NA	NA
	No	62	5								
	Yes	5	1								
Multifocal disease				1.05	0.19	5.94	0.70	0.07	0.00	1.36	0.08
	Single lesion	50	4								
	Multiple lesions	17	2								
Location (Tongue and mouth floor)				1.90	0.35	10.16	0.75	3.33	0.044	249.28	0.59
	No	47	3								
	Yes	20	3								
Location (Buccal and other sites except tongue and mouth floor)				0.67	0.11	4.20	0.97	1.43	0.013	152.81	0.88
	No	14	2								
	Yes	53	4								
Diabetes mellitus				1.90	0.28	13.04	0.87	NA	NA	NA	NA
	No	53	4								
	Yes	14	2								
Metformin treatment				0.70	0.10	4.78	0.90	0.60	0.033	11.05	0.73
	No	56	5								
	Yes	11	1								
Morphology				9.81	1.76	54.73	**0.03**	4.53	0.41	50.65	0.22
	Homogeneous	43	1								
	Non-homogeneous	24	5								
Pathology				7.30	1.21	44.06	0.09	6.07	0.33	112.57	0.23
	Low risk lesions †	50	2								
	High risk lesions ‡	17	4								
Recurrence				11.92	2.07	68.67	**0.02**	14.99	0.80	280.11	0.07
	No	48	1								
	Yes	19	5								
Area of oral leukoplakia (cm^2^)		2.58 ± 2.99	5.06 ± 6.31	NA	NA	NA	NA	1.04	0.71	1.54	0.83

Bold values denote statistically significant *p* value. * Four missing data of body mass index (n = 69). † Low risk lesions included squamous hyperplasia and mild dysplasia. ‡ High risk lesions included moderate and severe dysplasia. Abbreviation: NA, not available.

## Data Availability

Data sharing is not applicable to this article.

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
