# Peer review of "A Retrospective Cohort Study of Oral Leukoplakia in Female Patients—Analysis of Risk Factors Related to Treatment Outcomes"

_ijerph, 2021, doi:10.3390/ijerph18168319_

Round 1

Reviewer 1 Report

Oral Leukoplakia in Female Patients: Comparison with Male Patients and Analysis of Factors Related to Treatment Outcomes

In this retrospective study tried to evaluate the outcome laser therapy for oral leukolpakia, especially comparing differences between sexes. The study seems interesting but it presents many flaws which make it not suitable for publication in this present form.

  • Is an appropriate abstract included?

Yes, it is.

  • Is it clear what questions the paper is attempting to answer?

Yes, it is.

  • Are the objectives of the work clearly stated?

Yes, they are.

  • Are the methods clearly described?

  • No, the study methodology needs heavy implementation and revision: selection of participants, risk factors, PMD details must be better described: smoking habits and drinking habits must be specified; participant enrolment should better clarified: there is an enormous gap between number of males and females; the two groups should be more homogeneous and the different types of OL should be specified and considered in the analysis, being one of the most important factors for malignant transformation. Were all leukoplakias? Homogeneous for color and features?
  • In material and methods section, criteria for defining “current smoker” and “current drinker” must be defined.
  • Can you please clarify lines 59-61?
  • Why “margins involved by hyperkeratosis” were excluded?
  • Statistics need to be more detailed: study power? Sample size?

  • Does the paper provide new knowledge, either in the way of evidence or interpretation to what is already known in the field?

The methodology weakness unfortunately doesn’t allow any clear conclusion

  • Does the paper discuss an issue of current concern in the field?

Yes, it does

  • Is the paper suitable for an international readership?

No, the paper is not suitable for publication.

  • Are the arguments sound?

No, some revision should be done

  • Are the experimental data capable of supporting the conclusions drawn?

Probably, the differences between the number of patients included in the two groups are too important to draw definite conclusions

Reviewer 2 Report

[Suggestions]
The manuscript describes the cumulative malignant transformation rate (8.22%) of oral leukoplakia for female patients, and the area of oral leukoplakia to be the risk and prognostic factor related to postoperative recurrence. The findings of the manuscript may be interest to the readers in the Journal.

Minor:
L. 44: "200 mm2"
The authors need to check the description of the units.

L. 136, and 187: "Candida"
Candida should be in italics, just like in Tables 1, 3 and 4.

L. 146: "The ATR"
The ATR should be full spelled-out (probably Annual transformation rate).

L. 253: "National Health Service, Ministry of Health and Welfare"
It should be "Taiwan National Health ..." or so.

Reviewer 3 Report

This is an interesting retrospective work on the relationship between sex and oral leukoplakia

Some criticisms are present:

-The title does not appear centered, it is necessary to indicate the type of study and not to dwell in detail on the male / female relationship

-Remove all abbreviations from the abstract section

-Check that all keywords are Mesh terms Pubmed

-The Introdution section is too sparse and does not fully address the problem of malignant lesions of the oral cavity, neither as a clinic nor as therapeutic pathways, even if only general with particular reference to the most innovative strategies that can represent an element of attractiveness towards of scientific work

In this regard, I recommend that you insert the following scientific work in the reference section which could be of help:

-Lancellotta V, Pagano S, Tagliaferri L, Piergentini M, Ricci A, Montecchiani S, Saldi S, Chierchini S, Cianetti S, Valentini V, Kovács G, Aristei C. Individual 3-dimensional printed mold for treating hard palate carcinoma with brachytherapy : A clinical report. J Prosthet Dent. 2019 Apr; 121 (4): 690-693.

-Moreover, there is no modern classification of malignant lesions of the oral cavity which would certainly make the treatment of scientific work more complete. I recommend including in the reference section the following scientific work concerning even rarer forms of cancer:

-Negri P, Riccioni S, Lomurno G. Rare calcifying epithelial odontogenic tumor or Pindborg tumor. Case report and literature review [A rare odontogenic calcifying epithelial tumor, or Pindborg tumor. Report of a clinical case nd review of the literature]. Minerva Stomatol. 1999 Jul-Aug; 48 (7-8): 353-7. Italian. PMID: 10568113.

-At the end of the introduction section, insert the null hypotheses of the study

- Line 62 although briefly indicate the excision procedure

- When it comes to risk factors, which have also been recorded in the medical records, it is necessary to give more details on the degree of association reported in the literature.

-A paragraph indicating the limitations of the study is missing

Round 2

Reviewer 1 Report

A Retrospective Cohort Study of Oral Leukoplakia in Female Patients‒An Analysis of Risk Factors Related to Treatment Out- comes

Dear Editor,

        the authors highly improved the paper in order to make it clearer, nevertheless it is not suitable for publication in this present form, and in my opinion, could be misleading in some way.

General remarks: the study analyze the outcome of laser surgery on Oral leukoplakia, trying to point out which variables could influence the natural history on the lesions, especially gender. Data concerning males are not presented or presented in a very short way not allowing a correct comparison: to this regard, sample size was not calculated as the power of the study. Furthermore the study population seems highly unbalanced with a predominance of males. Of course, they present different risk factors and different transformation rate. This must be discussed. In general, the results are presented in a very chaotic writing, making the understanding of the clinical significance of the findings difficult for the reader. Its retrospective nature add few to the already known knowledge.

Some specific remarks are:

Lines 89-93. This represent a fundamental part of the study and it is not clear at all to me: incisional biopsies can be made either with blade or with laser device. Usually, for wide lesions, incisional biopsies are mandatory before performing excision. One of the most important parameter to start therapy is the presence of dysplasia which should diagnosed on the first incisional biopsy.

Line 95. It is hard to believe that the laser UltraPulse was the same across 20 years of study course. Can you give more details on that?

Line 108. Erythro -leukoplakias are usually classified within non-homogeneaous lesions. Were then excluded? Proliferative verrucous leukoplakias how were classified? Those have a complete different prognosis. And in those cases, did the authors remove all the pathological tissue?

Line 155. It is highly improbable that all the surgeries were uneventful, some minor complications should be listed.

Lines 186. Sample size and power of the study must be calculated.

Figure 1 is not readable.

Due to the above mentioned issues, the article is not suitable for publication.
